# Modulation of late Pleistocene ENSO strength by the tropical Pacific thermocline

Gerald T. Rustic [1,2 ✉], Pratigya J. Polissar [2,3], Ana Christina Ravelo[3] & Sarah M. White [4,5]

The El Niño Southern Oscillation (ENSO) is highly dependent on coupled atmosphere-ocean interactions and feedbacks, suggesting a tight relationship between ENSO strength and background climate conditions. However, the extent to which background climate state determines ENSO behavior remains in question. Here we present reconstructions of total variability and El Niño amplitude from individual foraminifera distributions at discrete time intervals over the past ~285,000 years across varying atmospheric $CO_2$ levels, global ice volume and sea level, and orbital insolation forcing. Our results show a strong correlation between eastern tropical Pacific Ocean mixed-layer thickness and both El Niño amplitude and central Pacific variability. This ENSO-thermocline relationship implicates upwelling feedbacks as the major factor controlling ENSO strength on millennial time scales. The primacy of the upwelling feedback in shaping ENSO behavior across many different background states suggests accurate quantification and modeling of this feedback is essential for predicting ENSO's behavior under future climate conditions.

[1] Department of Geology, School of Earth and Environment, Rowan University, 600 Whitney Ave, Glassboro, NJ 08028, USA. [2] Lamont-Doherty Earth Observatory, Columbia University, 61 Route 9W, Palisades, NY 19604, USA. [3] Ocean Sciences Department, University of California at Santa Cruz, Santa Cruz, CA 95064, USA. [4] Earth and Planetary Sciences Department, University of California at Santa Cruz, Santa Cruz, CA 95064, USA. [5] Present address: Department of Geography, University of California at Berkeley, Berkeley, CA 94720, USA. ✉email: rustic@rowan.edu

The tropical Pacific Ocean is home to coupled ocean-atmosphere interactions and feedback processes that result in local interannual sea surface temperature (SST) and wind field anomalies. These anomalies are the expression of the El Niño Southern Oscillation (ENSO), and they influence temperature and precipitation regimes across the globe[1]. Multi-model intercomparison efforts have displayed widely divergent ENSO responses to future climate conditions[2,3], largely due to differences in model expression of the feedback mechanisms that promote the growth and decay of the SST anomalies associated with ENSO[4]. The warmth and depth of the tropical thermocline influences the strength of many of these feedbacks by altering the surface temperature and vertical contrast of the eastern tropical Pacific where key ocean-atmosphere interactions occur[4–7]. These thermocline processes are dominant positive feedbacks in ENSO models[3,7,8], and thermocline conditions have been linked to ENSO activity in the Holocene[9] and the Last Glacial Maximum (LGM)[10]. ENSO's relationship with the background climate state, and therefore how it will change in the future, remains a central challenge in climate science.

Data and models support several hypotheses of the ENSO's relationship to background climate conditions. Evidence from many tropical climate archives, including corals, marine bivalves, lake and marine sediments, and individual foraminifera, have implicated insolation forcing as a key determinant of ENSO variability in the Holocene[9,11–15]. A synthesis of sub-annually resolved Holocene coral archives[16] and model simulations based on these data[17] suggest that internal variability dominates, however the short duration of these records may obscure long-term trends. ENSO activity has been linked to tropical Pacific mean conditions, including the East-West tropical SST gradient (E-W gradient), SSTs, and thermocline conditions, from sparse late-Pleistocene coral data[18]. Coupled ocean-atmosphere model simulations find that ENSO can be forced by insolation via alterations of the dynamical feedbacks critical for ENSO initiation and development. For example, higher insolation during the growth phase of El Niño events in boreal late summer/early fall leads to unequal zonal warming that alters the tropical Pacific wind fields[19,20], weakens the Walker circulation[21–23], and weakens the upwelling feedback by warming and/or deepening the tropical Pacific thermocline[4–7]. The tropical thermocline influences the SST anomalies in the eastern equatorial Pacific (EEP), altering zonal E-W SST and atmospheric pressure gradients[5,7,24], which in turn affect the wind fields that drive upwelling, setting up the ocean and atmospheric coupling important for ENSO[4]. These dynamical relationships may be altered during glacial states when lower sea level increases tropical land area and cooler SSTs reduce air-sea feedbacks[21,25]. Thus, both models and data suggest that the mean background state of the tropical Pacific Ocean plays a key role in shaping ENSO behavior, but the relative importance of SSTs, thermocline conditions, and insolation forcing remains unclear. Identifying linkages between tropical Pacific background conditions and ENSO variability may provide information on ENSO's relationship to large-scale climate and clues to future ENSO behavior.

Here, we reconstruct mixed-layer temperature variability and ENSO amplitude using populations of individual foraminifera from the central equatorial Pacific near the Line Islands spanning the past 285,000 years. We assess changes in total variability via statistical measures of population dispersion and also utilize Quantile–Quantile (Q–Q) analysis to determine specifically where population distributions differ. These targeted snapshots of central tropical Pacific oceanic conditions allow us to identify key relationships between El Niño amplitude and varying boundary conditions that test hypothesized links to ENSO behavior.

## Results

**Study location.** The data generated in this study derives from two sediment cores from near the Line Islands archipelago in the central equatorial Pacific (CEP). Core ML1208-17PC (hereafter 17PC) was recovered at 0.48°N, 156.45°W, at a depth of 2926 m, and core ML1208-14MC1 (hereafter 14MC) at 0.22°S, 155.96°W, from 3049 m water depth. This site lies within the NINO3.4 region and is near the location of Holocene coral[16,26] and individual foraminifera[9] ENSO reconstructions. Here, SSTs have a small seasonal cycle (±0.3 °C), but are highly variable at the interannual timescale (Fig. 1) with El Niño anomalies of +3.5 °C and La Niña anomalies of −2.8 °C. ENSO thus dominates mixed-layer temperature variability[9,27].

**Framework for characterizing ENSO change.** Distributions of individual foraminifera have been used to identify ENSO change in the central and eastern tropical Pacific in the Holocene and LGM[9,10,13,28]. The two- to four-week life span of each individual foraminifer captures monthly snapshots of ocean conditions, therefore analysis of multiple individuals provides a statistical sampling of mixed-layer conditions over the period of sediment accumulation. The total variability and shape of this distribution is determined by the annual cycle, interannual variability (including ENSO), and decadal and longer variability. Statistical modeling suggests individual foraminifera population variability is diagnostic of ENSO change in the CEP[27], and core-top calibration has demonstrated that individual foraminifera Mg/Ca captures population variability related to ENSO[29]. Quantile-Quantile (Q–Q) analysis has been used to assess ENSO change in individual foraminifera populations[9,10], and supported by with dispersion statistics provide a framework for determining ENSO change during specific sample intervals and under particular sets of climate boundary conditions.

Forward modeling of CEP mixed-layer temperatures was employed to characterize the changes in temperature distribution with changing variability parameters. Here we modified the SODA 2.1.6 35 m temperature (1958–2008)[30] to artificially alter the seasonal cycle, El Niño frequency, and El Niño amplitude both in concert and separately (Methods). These tests demonstrate that the warm and cold tails of the temperature distribution are most affected by ENSO change (Fig. 2), and that ENSO amplitude generates larger changes than the seasonal cycle or ENSO frequency in both statistical measures of temperature dispersion (e.g., standard deviation, variance, or median absolute deviation) and in the shape of temperature distribution. Simulated sampling of foraminifera from synthetic time series with ENSO change and with centennial-scale variability demonstrate that ENSO change is distinguishable and that long-term variability of the magnitude observed in coral records does not impact our interpretations (Supplementary Note 1). The response of temperature distributions at this location thus provides a useful framework for interpreting population changes: total dispersion is dominantly related to ENSO change (either frequency or amplitude), distinct changes in the population tails are influenced by ENSO amplitude, while the seasonal cycle has a minimal effect.

Our analysis of ENSO change for each sediment interval utilizes both Median Absolute Deviation (MAD), as well as changes in the tails of sample distributions identified using normalized quantile–quantile (Q–Q) plots[9,10] (See Methods). MAD change indicates alteration in total variability that, in the CEP, is highly dependent on ENSO as demonstrated by forward modeling the effects of ENSO change on temperature distributions (Fig. 2). Dispersion statistics such as MAD are unable to distinguish between changes in the warm (El Niño) or cold (La Niña) regions of the temperature distributions. Therefore, we

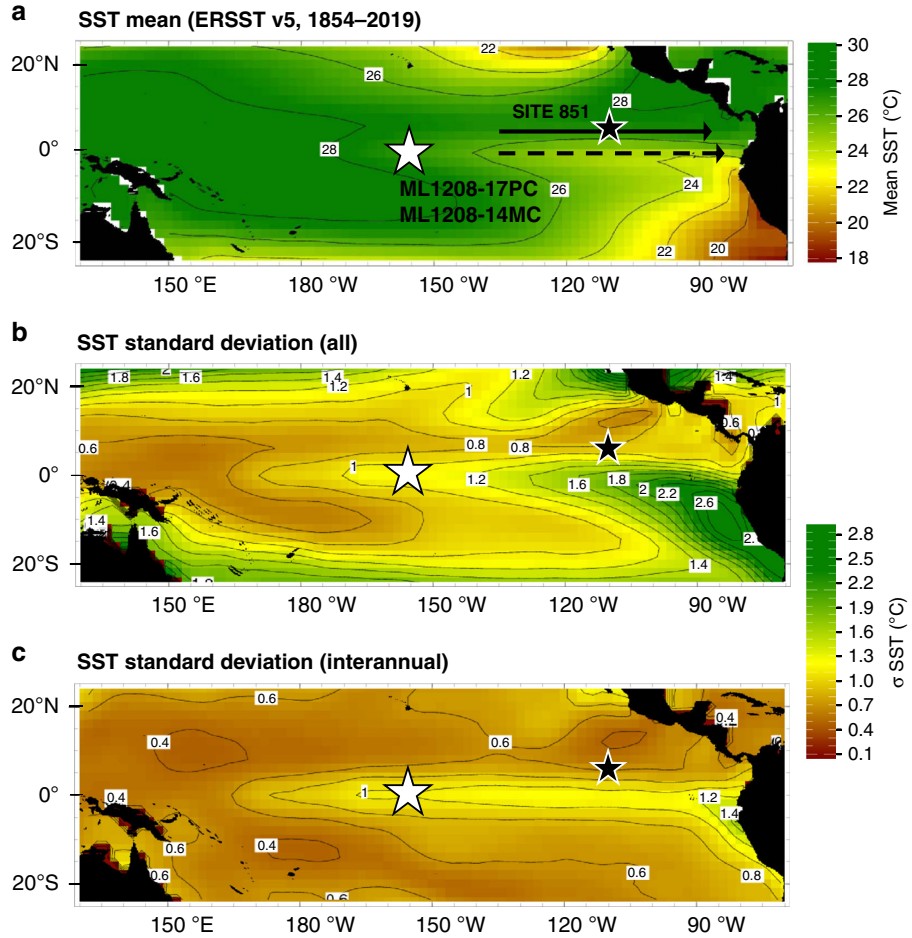

**Fig. 1 Map of the tropical Pacific Ocean showing the study site, sea surface temperature, and sea surface temperature variability. a** Average ERSST v5 sea surface temperature (SST) 1854–2019[52]. **b** Total SST variability as shown by the SST standard deviation. **c** Interannual variability, shown as the standard deviation of SST anomalies. Interannual variability is dominated by ENSO, thus variability at this site is largely determined by ENSO. The location of Line Islands cores 17PC and 14MC in the central tropical Pacific (this study) is shown as a white star with black outline; ODP Site 851 in the Eastern Tropical Pacific is shown as a black star with white outline. The solid black arrow in **a** schematically shows the mean location of the Equatorial Under Current. The dotted arrow depicts the location of the Equatorial Counter Current, whose depths are inversely related (based on ref. [36]).

characterize changes in El Niño amplitude, conservatively using only the warm end (90th–98th quantile) of Q–Q plots to reduce potential influence of depth habitat and changes in CEP subsurface structure. A significant change in El Niño amplitude is deemed to occur when the 90% confidence interval of any of these quantiles is below (above) the normalized 1:1 line (in normalized plots, the horizontal 0 line), indicating reduced (enhanced) El Niño amplitude. We then test these results for potential false positives (e.g., showing El Niño amplitude change where none has occurred) by comparing the number of significantly changed quantiles with randomly selected populations from our reference interval (Methods). Finally, we use the mean of the 94th–98th quantile as a single metric for El Niño amplitude change, as it shows a strong response to El Niño amplitude (Supplementary Note 1) and has a lower incidence of false positives than the 98th quantile alone. The common reference population for all samples is the CEP modern mixed layer temperatures15m–46,47m from SODA reanalysis data (1958–2009)[30].

**Mixed layer variability**. We measured Mg/Ca to determine mixed layer temperature from individual foraminifera at thirteen discreet intervals spanning 285,000 years and three glacial-interglacial cycles (See Table 1). Significant changes in total

population variability (Levene's test of absolute deviation, $p < 0.05$) are observed (Fig. 3), and we find that there are changes in total variability that exceed the uncertainty of our MAD measurements. The three intervals with lowest MAD values are observed at 111 ka, which is the glacial MIS 5d, and during interglacial periods at 197 ka and 230 ka. The three highest MAD values, whose lower uncertainty bounds exceed the upper uncertainties of the previous three intervals, are found at 127 ka, the warm interval of MIS5e, and in glacial intervals (MIS6) at 152 ka and 162 ka. As we find both relatively low and high MAD values during both glacial and interglacial periods, these changes in total variability appear unconnected to glacial state. Our forward modeling indicates that seasonal cycle alteration has little effect on MAD, and thus such large variability changes are likely indicative of differences in ENSO activity across these climate background states.

**Quantile–Quantile analysis**. We use Q–Q analysis of individual foraminifera populations from all interval to better characterize the observed changes in variability (Fig. 4). This analysis focuses on the warm temperature outliers that are characteristic of changes in El Niño amplitude. We find evidence of enhanced El Niño amplitude at 3.4 ka, 25 ka, 30 ka, 127 k, 152 ka, and 268 ka. Reduced El Niño amplitude is observed at 197 ka and 240 ka,

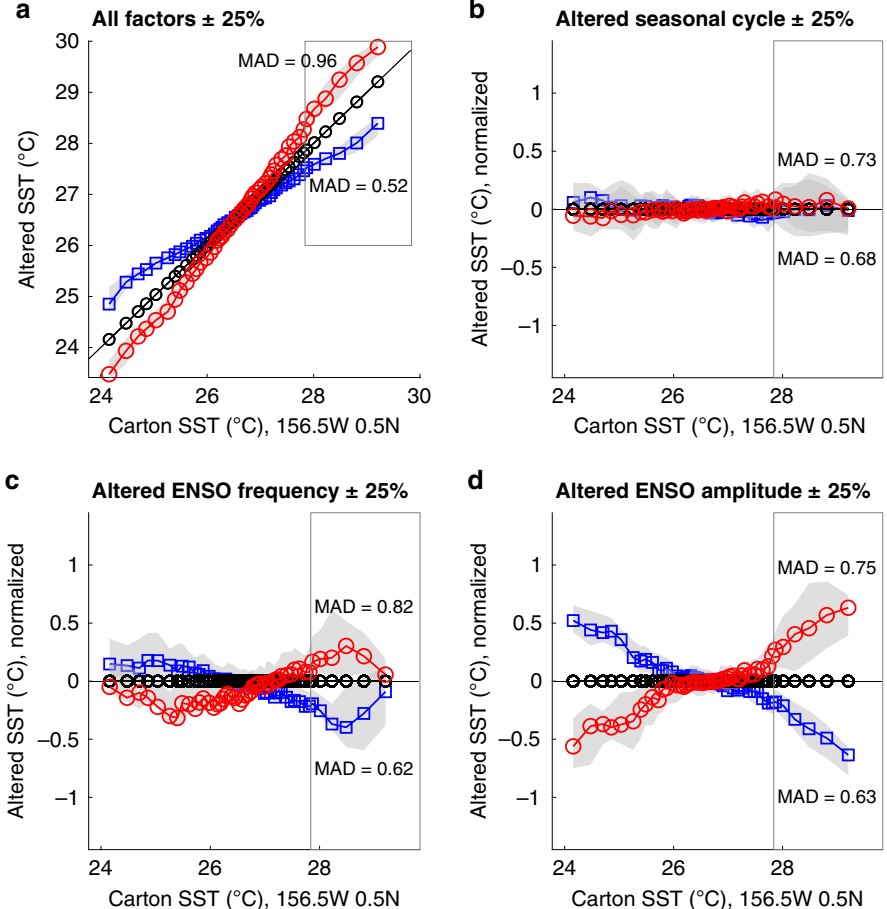

**Fig. 2 Quantile–quantile plots of modeled changes in mixed layer temperature with varying ENSO and seasonal cycle parameters at the Line Islands.**
Temperature data is from Simple Ocean Data Assimilation SODA 2.1.6 monthly reanalysis (1958–2008)[30] at 36 m for the grid box encompassing the core location. ENSO months are identified by National Weather Service Climate Prediction Center ONI values. The reference interval is the unchanged 36 m data set. Blue lines with square points show quantiles after a reduction in variability factors, red lines with circles show quantiles after increasing variability factors. The gray shaded regions show the 90% confidence interval of the quantiles of the altered data. The gray box outlines the ENSO-sensitive region of the population distributions (90th–98th quantiles). **a** Quantile–quantile (Q-Q) plot showing the changes in SST distributions resulting from a ±25% modification of the seasonal cycle, ENSO frequency and amplitude simultaneously. **b–d** Normalized Q-Q plots showing the decomposition of temperature changes under variation of one parameter: **b** ±25% alteration of the seasonal cycle, **c** ±25% change in ENSO frequency via addition or subtraction of ENSO events, and **d** ±25% change in ENSO amplitude via enhancement or reduction of temperature anomalies during ENSO events. The largest and most consistent changes in the ENSO-sensitive region are observed with changed ENSO amplitude. Median Absolute Deviation (MAD) is listed for each scenario showing the response of MAD to changing sources of variability. MAD changes are more pronounced under altered ENSO amplitude. MAD for the reference population is 0.71.

relative to modern mixed layer variability (15–47 m). While reduced El Niño amplitude is observed only during interglacial conditions (MIS7 intervals at 197 ka and 240 ka), enhanced El Niño amplitude is observed during both glacial and interglacial periods, and thus no consistent pattern emerges. Our finding of enhanced variability at 3.4 ka is consistent with the observed ~10% increases in EEP individual foraminifera δ18O variability at 3.2 ka[13]. We find enhanced El Niño amplitude prior to the LGM, when EEP individual foraminifera δ18O indicates higher variability[13] while Site 849 at the western edge of the EEP cold tongue experienced lower variability[10]. Our sample intervals precede these LGM samples by up to 2ky, so while direct comparison with these records is not possible, these results suggest possible shifts in El Niño behavior during the latter portion of MIS3 and the beginning of MIS2.

We assessed whether these findings were robust with respect to false positives via bootstrap analysis simulating individual foraminifera selection (Methods). We find that fewer than 3% of our simulations falsely report increased El Niño amplitude

when more than one ENSO-sensitive quantile shows statistically significant amplitude increase, and thus our findings of increased El Niño amplitude appear robust. The addition of analytical uncertainty to our temperature calculations leaves MAD values largely unchanged. The interval at 197ka shows significant El Niño amplitude reduction in only the 98th quantile, and our simulations show that over 20% of intervals with only this quantile reduced may represent a false positive result. Sample MAD in this interval is among the lowest measured in our study, and thus supports slightly reduced El Niño amplitude. As mixed layer variability has multiple sources and the quantile data is subject to considerable uncertainty, we urge caution interpreting this finding. The interval at 240 ka, however, has four reduced quantiles in the 90th–98th quantile band, which is found in fewer than 7% of our false positive tests. MAD values for this interval are among the lowest recorded in our sample populations, and thus both results are consistent with reduced El Niño amplitude. Higher MAD values, both from our sample populations and from simulations with added analytical uncertainty, are observed at

**Table 1 Summary of data from individual foraminifera analysis.**

| Interval Name | Core | Depth (cm) | Age (ky) | Mean $T$ (°C) | MAD | Q94-98T | $N$ | ENSO change |
|---|---|---|---|---|---|---|---|---|
| Mid-Holocene | 14MC | 4.5 | 3.4 | 28.18 | 1.00 | 0.56 | 150 | + |
| MIS 2 | 17PC | 60 | 25 | 26.5 | 0.96 | 1.04 | 65 | + |
| MIS 3 | 17PC | 72 | 29.6 | 26.38 | 1.08 | 0.61 | 82 | + |
| MIS 5d | 17PC | 286 | 111.8 | 27.1 | 0.75 | 0.30 | 63 | 0 |
| MIS 5e | 17PC | 322 | 126.9 | 28.12 | 1.14 | 0.74 | 77 | + |
| MIS 6.1 | 17PC | 378 | 152.2 | 26.7 | 1.21 | 0.57 | 79 | + |
| MIS 6.2 | 17PC | 400 | 162.2 | 27.18 | 1.08 | 0.41 | 79 | 0 |
| MIS 6.3 | 17PC | 442 | 181.3 | 27.05 | 1.00 | −0.02 | 80 | 0 |
| MIS 7a | 17PC | 475 | 197.1 | 28.04 | 0.83 | −0.17 | 82 | −* |
| MIS 7d | 17PC | 530 | 229.7 | 27.61 | 0.93 | 0.43 | 82 | 0 |
| MIS 7e | 17PC | 552 | 239.9 | 28.5 | 0.74 | −0.54 | 78 | − |
| MIS 8.1 | 17PC | 595 | 268.2 | 27.97 | 0.97 | 0.09 | 81 | + |
| MIS 8.2 | 17PC | 614 | 282.4 | 26.89 | 0.96 | 0.47 | 78 | 0 |

Columns are sample interval reference names and Marine Isotope Stage (MIS) names, core ID, sample depth (cm), sample age (ky), mean individual foraminifera temperature (°C), median absolute deviation (MAD), mean of the $94^{th}$–$98^{th}$ quantile normalized temperature (Q94-98T), number of foraminifera analyzed (N), and change in El Niño amplitude. For glacial periods MIS 6 and MIS 8, decimal notation refers to the indexed number of sampled intervals from that stage. El Niño amplitude change was determined by Q–Q analysis. Plus-signs indicate increased El Niño amplitude, minus signs reduced El Niño amplitude. The asterisk denotes an interval that displays significantly reduced El Niño amplitude at the 90% confidence level but may be a false positive result.

30 ka, 127 ka and 152 ka, intervals of highly likely enhanced El Niño amplitude. Thus, both total variability, as measured by MAD and quantile data strongly support the finding of increased El Niño amplitude at these times.

**Relationship with climate boundary conditions**. To determine whether ENSO change was related to changes in the background state of the tropical Pacific, we tested the relationship between independent reconstructions of climate boundary conditions and our reconstructed measures of ENSO change—total variability (MAD) and the mean normalized temperature anomaly of the $94^{th}$–$98^{th}$ quantiles (Q94-98T) (Fig. 5). These independent boundary condition reconstructions include: Insolation[31], a key control on ENSO in climate models; the position of the Intertropical Convergence Zone (ITCZ) as inferred from Ti in EEP ocean sediments[32], which influences the tropical Pacific wind fields; and the E-W SST gradient[33]. We interpolated values of these boundary conditions to our sample ages, and then calculated the correlation of these boundary condition reconstructions with our reconstructed measures of total variability (MAD) and El Niño amplitude (Q94-98T). Correlations were determined via linear fit modeling (e.g., simple linear correlation) using MATLAB R2019a and via bivariate weighted linear regression (WLR)[34] (see Methods). Using these correlation methods, we find no significant relationship between MAD or Q94-98T and the E-W SST gradient. Correlation of Q94-98T with insolation is high, as is the relationship between ITCZ position and both MAD and Q94-98T, however, these relationships are not significant at the 95% confidence level.

**Relationship with tropical Pacific thermocline**. To understand whether thermocline conditions influence ENSO, we compare both our CEP reconstructions of total variability (MAD) and El Niño amplitude (Q94-98T) with a reconstruction of EEP mixed layer depth inferred from the vertical contrast between shallow and deep dwelling foraminifera $\delta^{18}O$ at ODP Site 851 (3 °N, 111 °W). At EEP Site 851, the mixed-layer depth is indicative of the strength of the Equatorial Counter Current (ECC), responding to changes in the wind field of the eastern tropical Pacific. The mixed layer depth and strength of the ECC are inversely related to the mean depth of the EEP equatorial thermocline[35,36]. We find that MAD is significantly correlated with site 851 mixed layer depth using both linear fit modeling and weighted linear regression ($R = −0.58$, $p < 0.05$). We likewise find that El Niño

amplitude, as characterized by Q94-98T, is significantly correlated with the mixed-layer depth at site 851 ($R = −0.79$, $p < 0.05$) (Fig. 5). This finding links central equatorial Pacific variability, El Niño amplitude and the mixed-layer depth of the equatorial Pacific via the inverse relationship between EEP mixed layer depth at the equator and the mixed layer depth at site 851. Thus, periods of increased equatorial mixed layer depth, indicative of a warmer or deeper eastern equatorial Pacific thermocline, are associated with both reduced CEP variability and reduced El Niño amplitude. Conversely, periods of a reduced eastern Pacific mixed layer depth and a colder or shallower EEP thermocline are associated with increased CEP variability and El Niño amplitude. This result is consistent with model simulations demonstrating that warmer upwelling waters and a deeper thermocline reduce the ocean-atmosphere coupling in the eastern equatorial Pacific, weakening the feedbacks necessary for the growth of El Niño events[4,6–8]. Our results indicate that these feedbacks are important controls on the amplitude of El Niño events, and on overall ENSO variability, across a wide range of background climate conditions.

**Discussion**

That we find a significant relationship between CEP variability, El Niño amplitude and equatorial thermocline structure in the face of dramatically varying boundary conditions points towards the importance of the thermocline in controlling El Niño (and therefore, ENSO) expression across varying climate states. In the tropical Pacific, the SSTs and the surface wind field are tightly coupled. Reduced vertical contrast and a warming or deepening of the thermocline limits thermocline influence on the ocean surface, which in turn alters both surface and subsurface circulation[36]. Thus, a warmer thermocline reduces ocean-atmosphere coupling, limiting the influence of wind-driven upwelling and reducing the EEP temperature contrast both vertically and zonally. In the case of a warmer / deeper thermocline, the result is a dampening of the dynamical response of the EEP, and the feedbacks from these processes further deepen the thermocline and reduce upwelling, weakening their effect on ENSO growth. The primacy of thermocline and upwelling feedbacks in determining ENSO strength over long time periods has been found in simulations of ENSO change over the past 21ky[6], and over the past 300ky[7], but long-term reconstructions of ENSO have been largely limited to the Holocene and LGM (e.g., 9, 10, 13, 16, 28), with only limited data at few discrete time points from the last 130ky[15,18]. Our findings demonstrate that these processes are the

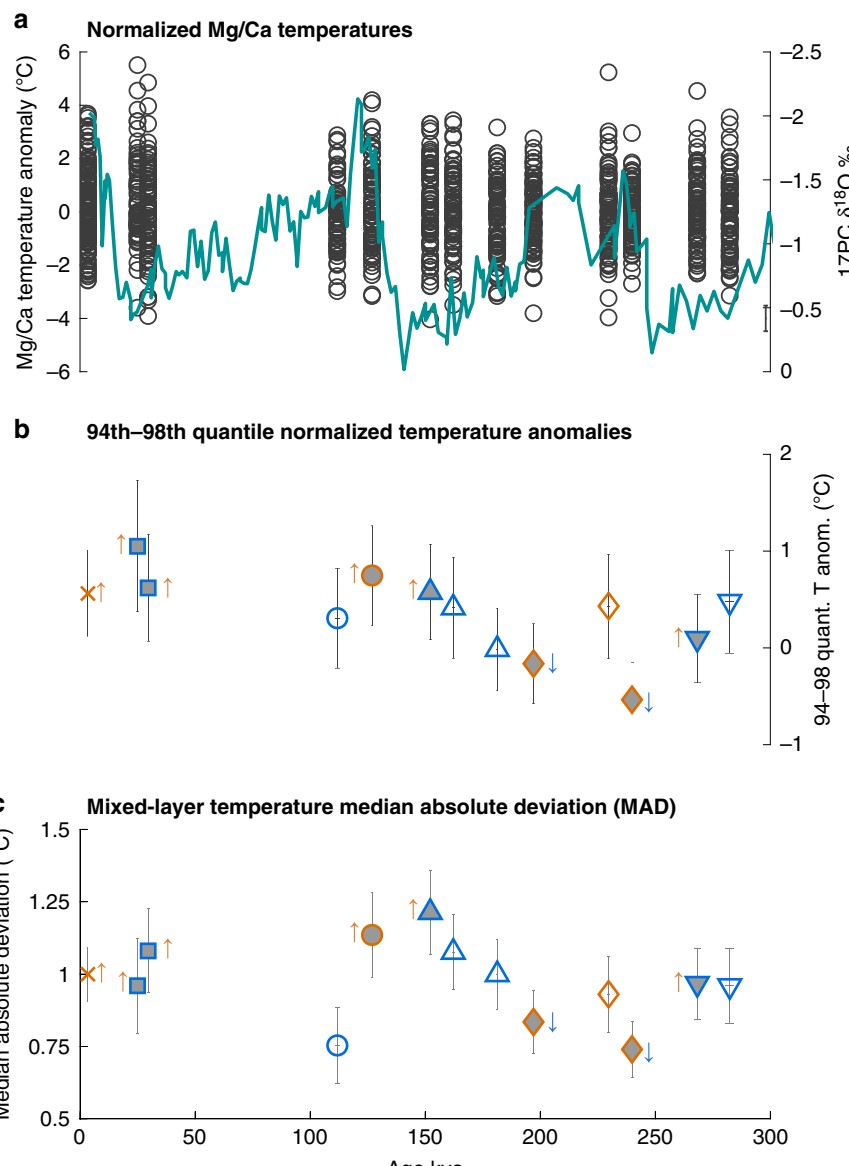

**Fig. 3 Results from analysis of individual G. sacculifer from Central Equatorial Pacific cores PC17 and 14MC. a** Mixed layer temperature distributions normalized to interval medians. Each circle represents one individual foraminifera Mg/Ca temperature. Average analytical error of ±0.47 °C is shown by the gray vertical bar on the lower right. The *Globigerinoides ruber* δ18O stratigraphy for core 17PC is displayed with these intervals showing their location on the chronology[53]. **b** Mean 94th–98th quantile normalized temperature anomaly. Positive anomalies suggest enhanced El Niño amplitude, while negative anomalies indicate reduced El Niño amplitude. Gray error bars represent the combined ~1-sigma resampling and analytical uncertainties. **c** Median absolute deviation (MAD) for each sample interval. Vertical whiskers show the standard error of the MAD. Symbols on **b** and **c** depict sample Marine Isotope Stages (MIS): Blue intervals are glacial periods, orange intervals are interglacials. Holocene, 'X'; MIS2-3, square; MIS5, circle; MIS6, triangle; MIS7, diamond; MIS8, inverted triangle. Filled symbols represent intervals that showed at least one ENSO-sensitive quantile suggesting either enhanced (up arrow) or reduced El Niño amplitude (down arrow) from quantile–quantile analysis.

best predictor of El Niño amplitude across widely varying climatic boundary conditions.

Modern data tightly links thermocline structure to the east-west zonal SST gradient, tropical wind fields, ITCZ position, and ENSO[1,5,37], but this relationship appears more complex across glacial-interglacial cycles (e.g., 21, 23, 25, 20). We find this complexity as well. It is possible that ocean circulation changes in response to exposure of the Sunda Shelf and/or alterations in the mean position and/or width of the ITCZ as a response to differing glacial hemispheric temperature gradients alters the Walker cir-culation intensity and spatial extent in ways not captured by static cross-basin measurements[25]. However, the observed changes in the eastern tropical Pacific thermocline as inferred from the

mixed-layer depth at site 851 are not clearly related to the reconstructed E-W SST gradient or mean ITCZ position as would be expected from modern data and theory, thus the origin of the thermocline signal is not clear from the paleoclimate data.

Model simulations and paleoclimate reconstructions provide clues to the possible origin of the thermocline signal. A Holocene El Niño amplitude reconstruction hypothesized that insolation-forced warming of the thermocline source waters warmed both the eastern and western equatorial Pacific thermocline, resulting in reduced El Niño amplitude[9]. Data show this reduction in El Niño amplitude was accompanied by warming subsurface tem-peratures across the equatorial Pacific from the Peru margin[38,39], to the EEP[24], and in the western equatorial Pacific[40,41]. At the

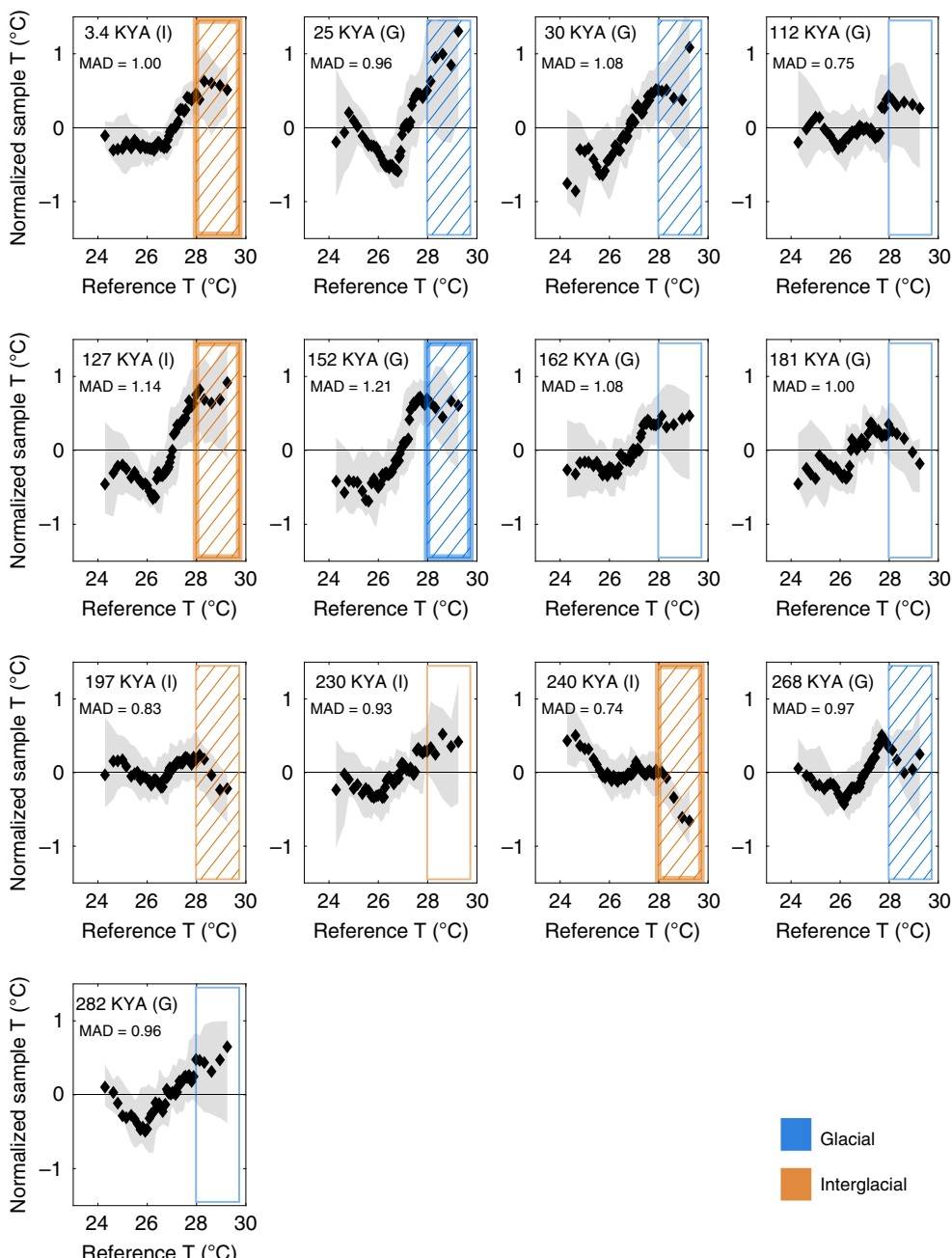

**Fig. 4 Normalized quantile–quantile diagrams for all sample intervals.** The reference interval is modern mixed layer temperatures (15–47 m) from Simple Ocean Data Assimilation SODA 2.1.6 reanalysis data set at 0.25°S, 155.75°W, 1958–2008, at the site of core 17PC[30]. Black dots represent quantiles 2–98, and gray shaded region is the 90% confidence region calculated from Monte Carlo resampling. The box outline indicates the region sensitive to El Niño amplitude change. Boxes with hatched fill display El Niño amplitude that differs from the reference interval in at least one of the 90th–98th quantiles at the 90% confidence level. Hatched boxes with heavier outlines have at least 3 of these quantiles that differ from the reference interval. Glacial-interglacial states are shown next to the age (G or I) and depicted by the box outlines (blue = glacial, orange = interglacial). Median absolute deviation (MAD) for each sample interval is listed. Non-normalized plots are found in Supplementary Fig. 9.

same time, warm SSTs are also observed near the region where equatorial thermocline waters are sourced by seasonal subduction in the Southern Hemisphere[42]. Comparable records from the thermocline source regions are lacking for the past 285,000 years. However, precessional forcing of extra-tropical Southern Hemisphere equatorial Pacific source waters has been linked to upwelling variations at site 1240 in the EEP cold tongue[43]. Southern Hemisphere insolation forcing is also reflected in the strength of the upwelling and associated feedbacks in the EEP with subsequent impacts on ENSO strength in model simulations

of the past 300,000 years[7]. Thus, both model results and palaeoceanographic data suggest that the tropical thermocline state is at least in part dependent on insolation signals transmitted to the EEP from the Southern Hemisphere via the source waters on millennial and glacial-interglacial time scales. A direct connection to insolation is not apparent in our data, which is not surprising given the complexity of these relationships. Regardless, our findings demonstrate that ENSO variability and El Niño amplitude are robustly linked to the tropical Pacific thermocline state across varying climate background conditions.

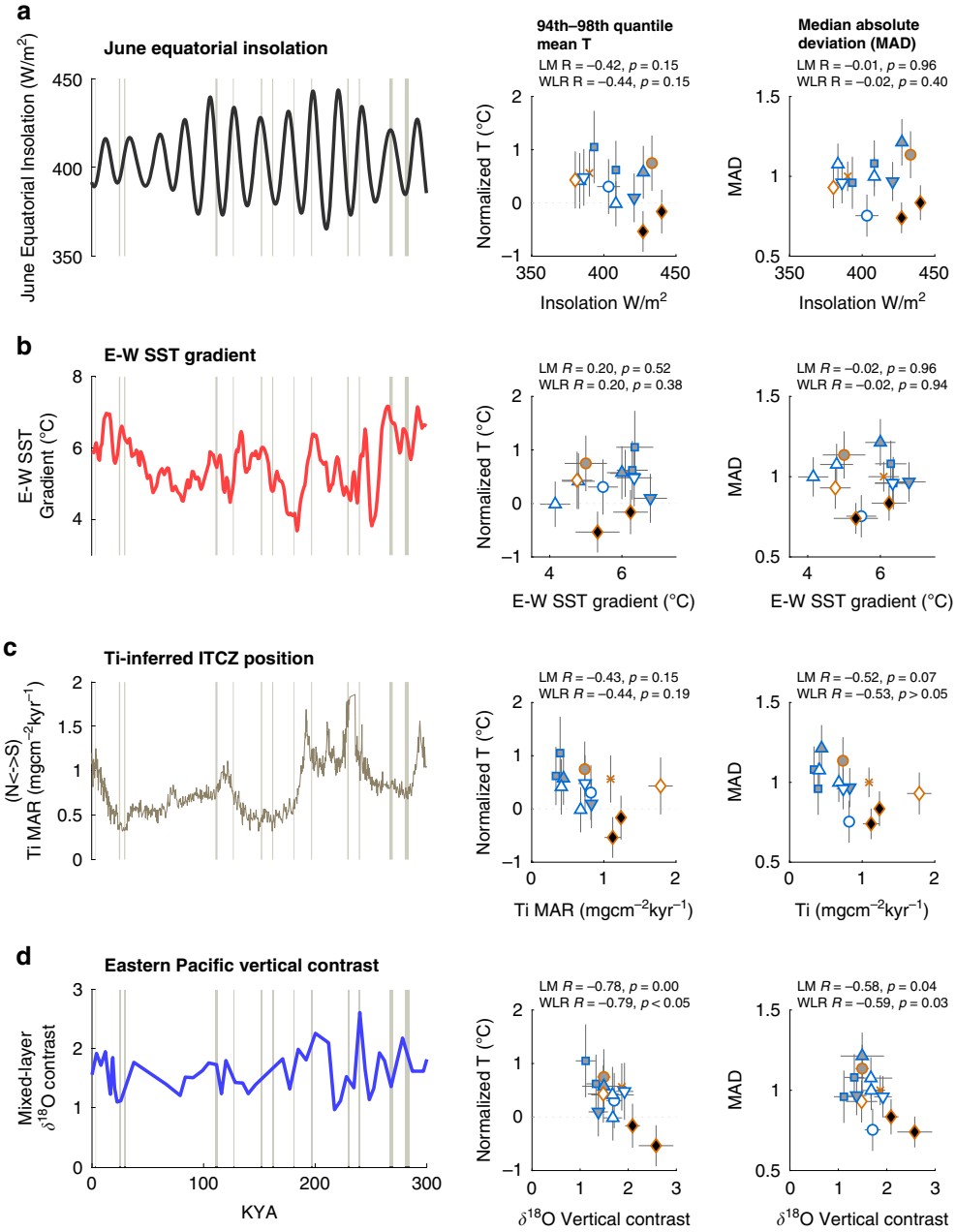

**Fig. 5 Relationship between hypothesized ENSO-sensitive background conditions and measures of ENSO activity.** Records of ENSO-sensitive background conditions are in the left column. Sample intervals are shown as vertical gray bars with widths proportional to the 1σ age uncertainty. Middle column shows the relationship between the climate background condition and the mean of the 94th-98th quantiles from our sample intervals. Right column shows the relationship between the climate background conditions and median absolute deviation (MAD) of calculated temperatures. In the right two columns vertical gray bars represent the ~1-sigma uncertainty in the mean of the 94th-98th quantile (middle) and standard error of the MAD (right). Gray horizontal bars depict the age-related uncertainty in the climate background conditions. For the correlation plots, the mid-Holocene interval is shown as an X, Marine Isotope Stage (MIS) 2–3 intervals are squares, MIS 5 intervals are circles, MIS 6 intervals are triangles, MIS 7 intervals are diamonds, and MIS 8 intervals are inverted triangles as in Fig. 3. Glacial intervals are depicted in blue, interglacial intervals are in orange. Filled markers represent intervals with altered El Niño amplitude significant at the 90% confidence level (black) or increased El Niño amplitude (gray). Correlation coefficients (R value) and statistical significance from a linear fit model (LM) and weighted bivariate linear regression (WLR) between each climate parameter and the measures of ENSO activity are shown. **a** June insolation at 0°N[31]. **b** Multi-proxy tropical Pacific East-West SST gradient[33]; **c** Inferred position of the intertropical convergence zone (ITCZ) from Titanium Mass Accumulation Rate (MAR)[32]; **d** Vertical contrast (inversely related to mixed layer depth) from ODP Site 851 in the Eastern Pacific. Vertical contrast is calculated as the difference between *G. sacculifer* and *Globorotalia tumida* δ18O[36].

Our reconstruction of ENSO at discrete time intervals over the past ~285,000 years demonstrates a significant relationship between El Niño amplitude and the warmth/depth of the equatorial thermocline. Periods with a warmer/deeper equatorial thermocline exhibit reduced El Niño amplitude, while shoaling of

the thermocline is associated with enhanced El Niño amplitude. We find that the overall variability of the CEP as measured by the median absolute deviation (MAD) also has the same relationship. As total variability in the CEP is dominated by ENSO and our quantile records reconstruct El Niño amplitude, together these

findings provide direct connections between ENSO strength and eastern tropical Pacific dynamics, implicating upwelling and thermocline feedbacks as the key mechanisms controlling ENSO expression on glacial-interglacial time-scales. The primacy of these feedbacks versus other hypothesized influences is demonstrated by the consistency of the ENSO-thermocline relationship across widely disparate climate boundary states. The main source of divergent predictions of ENSO behavior under future warming scenarios is the thermocline feedback[4], and thus our results provide an important paleoclimate context for such predictions. While the response of the climate system to future greenhouse gas forcing need not have the same expression as glacial-interglacial forcing, our results reinforce the importance of the thermocline in shaping ENSO behavior under varied climate states.

## Methods

**Age model**. The age for the sample interval from core 14MC is from a previously published radiocarbon-based age model[9]. The age model for core 17PC was re-tuned to the LR04 benthic $\delta^{18}O$ stack[44] using the original *Globigerinoides ruber* $\delta^{18}O$ stratigraphy via multiple resampling and averaging of nine age control tie points to better characterize the major transitions observed in the record and to align it in its entirety to the LR04 stack. The new age model is within 2.1ky of previous age models for the last 150ky (Supplementary Note 2).

**Analytical methodology**. Near-surface ocean temperature variability was reconstructed by analysis of elemental Mg/Ca ratios on individual specimens of the mixed-layer dwelling foraminifer *Globigerinoides sacculifer* (without the final sac). From each sediment interval, 63-150 individuals were analyzed to provide statistically robust sampling. Each one-cm sample represents ~400–600 years based upon the average sedimentation rate of 2.2 cm/ky over the last 285ky in core 17PC, with minimum sedimentation rates of 1.35 cm/ky. At typical bioturbation depths of 5 cm, each sample integrates ~1000–2500 years, providing millennial-scale resolution.

**LA-ICPMS analysis**. Individual specimens of *G. sacculifer* were analyzed for trace metals via laser ablation inductively coupled mass spectrometry (LA-ICPMS) following the protocol outlined by Sadekov et al.[24] and detailed by White et al.[9] Individuals were selected from the 355–425 μm size fraction to reduce ontogenetic effects[45]. Each individual foraminifer was sonicated in deionized water and washed with methanol or ethanol. LA-ICPMS trace metal analysis was performed on a Photon Machines Analyte.193 with HelEx sample cell with a Thermo ElementXS inductively coupled plasma mass spectrometer. Previous LA-ICPMS reconstructions analyzed the final chamber from the inside out[9,10,46]. Here we ablated from the outside of the test toward the inside (Supplementary Fig. 8) and on multiple growth chambers. Three 50μm diameter targets were ablated on the final chamber (f0) of each individual, and two were ablated on the second-to-last chamber (f1). Elemental abundances ($^{11}B$, $^{24}Mg$, $^{25}Mg$, $^{27}Al$, $^{43}Ca$, $^{44}Ca$, $^{55}Mn$, $^{66}Zn$, $^{88}Sr$) were measured and an average Mg/Ca ratio was calculated for each target. A whole chamber Mg/Ca value was calculated as the average of all targets on a given chamber. A whole foraminifer Mg/Ca value was calculated via weighted average of the f0 (55%) and f1(45%) chambers based upon sequential analysis of individual foraminifera, first by LA-ICPMS, then by traditional cleaning and dissolution ICP-OES. These weights give the maximum correlation between the two methods.

**Conversion of Mg/Ca ratios to temperature**. Mg/Ca ratios from individual foraminifera were converted to temperature by first applying a multispecies dissolution correction[47] using modern oceanographic data from CLIVAR P16 at 0° and 1° N, 151° W[48] to calculate values of $\Delta[CO_3^{2-}]$ (1.94 μmol/kg) using CO2calc software[49]. Temperature was calculated from resulting Mg/Ca ratios using the multispecies calibration equation $Mg/Ca = 0.38 \pm 0.02 \times exp(0.090 \pm 0.003 \times T)$, where $T$ is temperature in Celsius[50].

**Quantile–Quantile analysis**. We use quantile–quantile (Q–Q) analysis to determine whether the changes in individual foraminifera distributions are the result of ENSO change by identifying the portions of our sample distributions that differ from a reference interval. In this analysis, quantiles of a sample population are compared to quantiles of a reference population. Based on forward modeling, changes in ENSO parameters are observed as differences in the warm and cold tails of our sample distributions (Supplementary Fig. 3, Supplementary Note 1). We calculate the quantiles of our distributions based on the method of Ford et al.[10] and White et al.[9], which is summarized here. Individual foraminifera temperatures are used to calculate an empirical cumulative density function (ECDF) for each sample interval that is then used to calculate the quantiles of that temperature distribution at 2% intervals. We resample (with replacement) along this ECDF 10,000 times to calculate our confidence intervals (90%). This approach assumes our sample

distribution is a reasonable representation of the true distribution and that the differences between the true distribution and our ECDF are contained within our error estimates generated by bootstrap resampling. This method does not generate new or hypothetical data, and is thus limited by the range of the sample population. This method follows that outlined by Press et al.[51], slightly modified to use a continuous ECDF rather than discrete resampling. In this analysis, we assume that our sample data is a close approximation of the actual data, and that the actual data is likely represented within our confidence intervals. Bootstrap confidence intervals may differ from those calculated via other means, but conform to the statistical principals outlined by Press et al., and have been previously used to assess foraminiferal populations[9,10]. Sample quantiles are compared to corresponding quantiles of the common reference population by plotting sample quantiles and confidence intervals on the y-axis and the reference quantiles on the x-axis. Identical distributions will plot along a 1:1 line, and distributions that differ in mean fall along an offset line with slope equal to one. Divergence from a line with slope equal to one shows the portions of the distribution where the populations differ. We normalize plots for further analysis by subtracting a line with slope 1 that passes through the mean of the sample distribution, removing the effect of mean offsets in temperature (such as from glacial cooling). We use the SODA 2.1.6 1958–2008 mixed layer temperature (15–46 m) from the Central Equatorial Pacific 0.5°x0.5° grid box surrounding site 17PC as the common reference interval for all of our sample comparisons and for computing the normalized quantile temperature anomalies. These are calculated as the difference between the normalized reference temperature (zero) and the temperature of the quantile in question.

**False positive test**. We assessed the likelihood of falsely identifying ENSO change, specifically changes in El Niño amplitude, via Monte Carlo resampling. We generated 1000 synthetic populations from our reference population by selecting 80 monthly temperatures values (representative of the number of individuals foraminifera sampled in an interval) and applying random analytical uncertainty to simulate foraminifera temperature values. Q–Q analysis was performed on each population with the original mixed-layer temperature record as the reference population to determine whether the synthetic populations showed statistically significant change in the 90th–98th normalized quantiles characteristic of El Niño amplitude change, and how many quantiles were affected. We find that false positives for increased El Niño amplitude are rare – fewer than 5% of the synthetic populations displayed a significant increase any of the 90th–98th quantiles, and fewer than 3% displayed two significantly increased quantiles. Thus, it is unlikely that a sample showing increased El Niño amplitude comes from a population with the same distribution as the reference interval. We find that ~20% of the synthetic populations contain at least one of the 90th–98th quantiles with a significant El Niño amplitude reduction, but fewer than 7% of synthetic populations display a reduction in three or more of the 90th–98th quantiles. Thus, robust identification of reduced El Niño amplitude requires multiple quantiles in the 90th–98th range to exhibit a statistically significant reduction.

**Correlation with climate parameters**. We assessed the relationships between independent climate parameters and the measures of ENSO activity calculated in our study using linear modeling ('fitlm') in MATLAB 2019a and multiple axis reduction bivariate weighted linear regression (WLR[34]), which considers uncertainties (specified as the standard error) in both the reconstructed climate parameters and our ENSO activity variables. We generated values from the reconstructions of climate parameters via linear interpolation at our sample ages. We generated uncertainty estimates for these values using Monte Carlo methods that incorporate age and analytical uncertainties from those climate parameters and age uncertainties in our age model (See Supplementary Note 3). Multiple ($10^3$) age estimates for each sample interval were generated for each independent climate record using the combined age uncertainty of record and our age model. We then re-interpolated each independent climate parameter at the generated ages incorporating analytical uncertainty from these records. Total uncertainty for each reconstructed parameter was calculated from the empirical bootstrap results as one half of the 2.5–97.5 quantiles range to approximate one standard deviation of the resulting data distributions. Additional uncertainty was added to the interpolated values of the comparison climate record from site 851 for intervals with data points outside of our combined age error estimates. We based these additional errors on the amplitude of the dominant period of this record (obliquity) and the proportion of that period between our ages and the nearest data point (Supplementary Note 3). We then determined the correlation coefficients and statistical significance of the correlations between each calculated climate parameter and the Median Absolute Deviation (MAD) and mean of the 94th–98th quantiles (Q94-98T) derived from our central Pacific mixed layer SST data. We use MAD as a robust measure of dispersion calculated from the median value of population residuals. This measure is less sensitive to outliers (which is where much of the ENSO variability is found), but incorporates all sources of variability, including annual, interannual, decadal and longer change, and does not differentiate between the warm (El Niño) and cold (La Niña) tails of the distributions. However, today the total variability signal, and hence MAD, is dominated by ENSO variability in the CEP, and thus this measure of total variability is indicative of ENSO (Figs. 1 and 2). We estimate the uncertainty of our MAD parameter as the standard error of the MAD. The significance of our results is not altered by using the variance or standard deviation of the

calculated interval temperatures. We use Q94-98T to characterize the response of the warmest tail of the temperature distribution to ENSO change (Fig. 2). This tail is particularly sensitive to El Niño amplitude, and thus, while related to MAD, provides additional information regarding the source of observed variability change. Uncertainty in the 94th–98th quantiles for bivariate regression was calculated from the distribution of 94th–98th quantile means generated from the Monte Carlo resampling performed to generate the Q–Q analysis. As with the climate parameters, we use one half of the 2.5–97.5 quantile range to approximate one standard deviation of the resulting data distributions. This method provides uncertainty estimates comparable to the standard deviation, while accounting for the possible non-normality of our results. Analytical uncertainty is added in quadrature to determine the total uncertainty about the estimate of the mean.

## Data availability

Individual foraminifera data presented here is available via National Oceanic and Atmospheric Administration National Climatic Data Center archive at https://www.ncdc.noaa.gov/paleo/study/26871.

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

## Acknowledgements

The authors would like to thank Jean Lynch-Stieglitz and the MGL1208 shipboard party, and Rob Franks at UCSC for analytical assistance. Funding was provided by National Science Foundation grants OCE-1401649 and OCE-1405178.

## Author contributions

G.T.R., P.J.P., and A.C.R. designed the research. G.T.R. and S.M.W. performed geochemical analysis. G.T.R. and P.J.P. analyzed the data. G.T.R., P.J.P., A.C.R. and S.M.W. contributed to the preparation of the manuscript.

## Competing interests

The authors declare no competing interests.
