## [Peer Review File · Nature Communications]

Reviewers' Comments:

Reviewer #1:

Remarks to the Author:

The study By Rustic et al. attempts to reconstruct ENSO variability and El Nino amplitude using the median absolute deviation (MAD) and 94th-98th percentile of distributions of single foraminifera Mg/Ca measurements respectively. Based on 13 intervals across various background conditions, the authors conclude that upwelling feedbacks played a major role in controlling ENSO behavior.

I appreciate the author's efforts to include different statistical measures to ensure that their conclusions are not a direct consequences of their choice of statistics. Although I remained unconvinced that using the upper quantiles is a sound approach, I appreciate the inclusion of the robust measure of MAD, and I would encourage the authors to focus on this measure instead in resubmission.

Another suggestion that may improve on readability would be to include the analytical uncertainty on some of the figures. The foraminifera Mg/Ca are not actual single shell measurement, making it more difficulty to consider any of these points (including in the tails) measurement errors only (i.e. not 63 data points).

Pending these two comments and the minor editorial details below, I'd recommend for publication.

Minor comments:

Figure 3: It's extremely confusing to use the same colors to represent different conditions (glacial/interglacial and ENSO sensitivity) using outlines and filled. I'd suggest using different colors here (yes, I understand blue/red is temperature).

Supplementary information: references appear as "error, reference source not found" on the PDF.

Reviewer #3:

Remarks to the Author:

This paper provides new evidence for the impact of background climate conditions on ENSO frequency and amplitude. The novel result of this paper is to demonstrate a relationship between the intensity of the upwelling, the thermocline depth and the mixed layer depth in the Central Pacific and the amplitude frequency of past ENSO events. Evidence is based on data obtained from a novel approach using single foraminifera analyses. Multiple measurements of planktonic foraminifera (*G. sacculifer*) based temperatures derived from discreet layers of sediment sequence (corresponding to 100's of years of sedimentary deposit) enable to estimate ENSO amplitude and strength.

The statistical processing and significance of this data is discussed in detail by the authors in the body of the paper and in the responses to comments of three reviewers. I am not an expert with the statistical processing, but I find the response of the authors to be detailed, robust and convincing. The use of single foraminifera data in this field is novel, however there are previously published papers,

quoted within this paper, that provide details of the methods for readers who wish to obtain more detailed information. From the response to the reviewers and authors comments my assessment is that a very thorough processing of the data has been conducted.

I have 3 concern.

- Firstly, the modern mixed layer temperature indicated as the thin dotted line on Fig.3 provides the baseline for what the authors thereafter call increased decrease variability/amplitude of ENSO. It seems critical to add an error bar to this value, as done for the estimates for past MAD's. I would also welcome an indication of the MAD of the modern mixed layer depth temperatures for typical El Niño Years and La Niña years. These values should/could be indicated on Fig.3? This would enable to quantify past shifts in ENSO and avoid the use of "stronger" versus "weaker" ENSO.

- Secondly, I would like to see some discussion on the chronology. If the authors were to plot the foraminifera del-18O of core PC17/14MC1 in the background of figure 3 (bottom panel) this would enable the reader to visualise the chronology. If this is too crowded then at least as supplementary material. There is a lot of variability within MIS so simply quoting these are MIS7 or MIS5 samples does not enable the reader to clearly associate them with warm/cold periods.

- Finally, the comparison with the downcore mixed layer del-18O contrast data (Fig 5 d). I cannot see the exact resolution of the data but it is clearly much lower than the Ti data (Fig 5c). The grey shaded bars highlight the periods when there is a direct comparison of the single foraminifera data and downcore core data. Some of these periods are located at core depths where there seems to be no data as for example between 150-170ka (the authors should clarify this). If this is the case then these data should be excluded from the right hand plots. This may not change the relationship significantly but would be more robust.

TITLE/ABSTRACT: The title refers to changes in the thermocline yet there is no mention of thermocline in the abstract that refers to mixed layer depth and upwelling... May I suggest either a change of title to links with mixed layer depths or indicate that mixed-layer depth and thermocline are related in the abstract?

OVERALL ASSESSMENT: I strongly believe that this is a novel and truly thought provoking paper with a new set of data much needed. This paper will be followed by model data inter-comparison and provides a hypothesis to test observed differences in current models predicting future ENSO variability. These models often differ in their conclusions and perhaps this is related to the way in which they integrate changes in the ocean's thermal structure.

Details:

- can't see the grey vertical shaded bars on fig 5.
- could cite the Regoli et al paper 2014 which provides a detailed reconstruction of thermocline depth (although in the Western pacific)
- Line 98 : reword to : but are highly variable at interannual timescale...
- Line 103-104: what do you mean by artificially alter the seasonal cycle?... maybe to be added in sup mat.
- Line 225, I would suggest adding the Driscoll et al., 2015 paper that is the only one to provide quantitative measures of Glacial ENSO variance.
- Lines 250-262 there is a bit of repetition in this final paragraph.

Reviewer #4:
Remarks to the Author:

Review of Rustic et al., NCOMMS-20-04764

Gerald Rustic and coauthors provide a new reconstruction of ENSO amplitude for 13 time intervals over the last 300,000 years, based on individual foraminifera temperatures in a sediment core in the tropical Pacific. They base this reconstruction on changes in the probability density functions (PDFs) of Magnesium/Calcium ratios. A pseudoproxy method is devised to connect sea surface temperatures, calcification temperatures of individual foraminifera and ENSO characteristics (amplitude, frequency, seasonality) from observational 2D temperatures in the core gridbox. They go on to show fairly constant mean temperatures for the 13 time periods, but suggest that the mixed-layer temperature median absolute deviation (MAD) varies considerably. Based on quantile-quantile plots they suggest that the upper tail of the distribution varies in the different time periods, and this is linked to altered ENSO amplitude in their simplified model. Correlation of the estimated temperatures for the individual foram. distributions suggests a link to the thermocline depth (via the difference in $\delta^{18}O_{calc}$ for two species, taken as proxies for the mixed-layer $\delta^{18}O$ contrast and, consequently, for thermocline depth). Several hypotheses around the correlation between orbital forcing, thermocline depth and ENSO amplitude are discussed, but no firm conclusion is reached.

The dataset is clearly interesting, and the analysis novel. I am not convinced, however, by the results and the discussion.

GENERAL COMMENTS:

* Technical comments: The figures are not colorblind friendly and have a very nonlinear color scale exaggerating contrasts (Fig.1). There are unresolved references in the supplementary, obscuring the technical derivation of the pseudoproxy/statistical model.

* Some of the arguments appear circular: Reconstructing the thermocline temperature variability the authors suggest that thermocline variability is feeding back into ENSO - but both reconstructions are based on related data, it appears, and in any ENSO change we would expect a thermocline temperature change.

* The comparisons (to individual records for the ITCZ position/ zonal temperature gradient) are not ideal to allow to draw firm conclusions on the presence or absence of explanatory feedbacks for the observed scatter in reconstructed ENSO amplitude, in particular as uncertainties (age, measurement, calibration) are not taken into account.

* The use of individual foraminifera analyses, and the methodology put forward, is interesting and the aims of the study are important.

SPECIFIC COMMENTS:

- L19: It is unclear from the discussion, and the, what the climatic state here actually refers to - the global climate (glacial vs. interglacial) or the equatorial Pacific. Below it is highlighted that the results indicate no sensitivity to the climate state. This is one of the key findings, and not surprising, given e.g. what we learn about ENSO from the mid-Holocene CMIP model intercomparison.

- L23-25: Isn't it obvious that ENSO variability is highly correlated to upwelling/thermocline depth? The "upwelling feedbacks" remain vague here and in the discussion (page 11/12).
- L93: While age uncertainty is not impacting the PDFs, it will affect the correlations to the mixed-layer d18O contrast (Fig.5), and this is ignored. Also, it is unclear (caption of Fig.5) what the actual shown age uncertainty refers to (1 sigma? 2 sigma? quantiles?). This information would help to judge the robustness of the results.
- L101: The 'calibration' of the forward model centers on the variables 'seasonality', 'event frequency' and 'amplitude' - but what about spatial expression? A wide range of ENSO flavors have been suggested, and changes in the spatial warming structure may well obscure results obtained from a single location.
- L258: typo 'hypothesis'
- Figure 3: Given that these errors are 1 MAD, all these estimates are actually not statistically different.

Reviewers' Comments:

Reviewer #3:

Remarks to the Author:

The revised version of Rustic et al., has been greatly improved, brings novel ideas and concepts on the relationship between interannual climate variability and background climate state. The authors have addressed the questions posed by myself and other reviewers. They have a very novel data set and have done a thorough statistical analysis of their data. Although some questions remain about the lower resolution of the Pacific vertical contrast record used the methods used by the authors to process and compare the data sets is thorough. I only have some minor comments. I recommend publication.

Mary Elliot

Line 51: change mussels to marine bivalves

Line 53 add the Driscoll et al paper as showing the role of insolation.

Caption for Figure 3 could be simplified to : ... Vertical whiskers show the standard error of the MAD. Symbols on b and c depict sample Marine Isotope Stages (MIS): Blue intervals are glacial periods, orange intervals are interglacials. Holocene, 'X'; MIS2-3, square; MIS5, circle; MIS6, triangle; MIS7, diamond; MIS8, inverted triangle.

Reviewer #4:

Remarks to the Author:

The provided revised manuscript is much improved. My comments have been addressed satisfactorily.

Reviewers' Comments:

Reviewer #3:

Remarks to the Author:

The revised version of Rustic et al., has been greatly improved, brings novel ideas and concepts on the relationship between interannual climate variability and background climate state. The authors have addressed the questions posed by myself and other reviewers. They have a very novel data set and have done a thorough statistical analysis of their data. Although some questions remain about the lower resolution of the Pacific vertical contrast record used the methods used by the authors to process and compare the data sets is thorough. I only have some minor comments. I recommend publication.

Mary Elliot

Line 51: change mussels to marine bivalves

Line 53 add the Driscoll et al paper as showing the role of insolation.

Caption for Figure 3 could be simplified to : ... Vertical whiskers show the standard error of the MAD. Symbols on b and c depict sample Marine Isotope Stages (MIS): Blue intervals are glacial periods, orange intervals are interglacials. Holocene, 'X'; MIS2-3, square; MIS5, circle; MIS6, triangle; MIS7, diamond; MIS8, inverted triangle.

Reviewer #4:

Remarks to the Author:

The provided revised manuscript is much improved. My comments have been addressed satisfactorily.

Reviewer comment and revisions:

The revised version of Rustic et al., has been greatly improved, brings novel ideas and concepts on the relationship between interannual climate variability and background climate state. The authors have addressed the questions posed by myself and other reviewers. They have a very novel data set and have done a thorough statistical analysis of their data. Although some questions remain about the lower resolution of the Pacific vertical contrast record used the methods used by the authors to process and compare the data sets is thorough. I only have some minor comments. I recommend publication.

Mary Elliot

- We thank Dr. Elliot for the review comments.

Line 51: change mussels to marine bivalves

- We have changed this to the more appropriate term.

Line 53 add the Driscoll et al paper as showing the role of insolation.

- We have added this reference (making the above edit very appropriate) and have updated the reference list.

Caption for Figure 3 could be simplified to : ... Vertical whiskers show the standard error of the MAD. Symbols on b and c depict sample Marine Isotope Stages (MIS): Blue intervals are glacial periods, orange intervals are interglacials. Holocene, 'X'; MIS2-3, square; MIS5, circle; MIS6, triangle; MIS7, diamond; MIS8, inverted triangle.

- This edit has been performed to simplify the caption, and improves readability.

Reviewer #4 (Remarks to the Author):

The provided revised manuscript is much improved. My comments have been addressed satisfactorily.

- No changes made. We thank reviewer 4 for the constructive comments on previous reviews.